# Workplace Mental Health Status Among Academic Staff: Psychological Distress, Burnout, and Organisational Culture at a South African University

**DOI:** 10.3390/bs15101410

**Published:** 2025-10-16

**Authors:** Veena Abraham, Johanna C. Meyer, Kebogile Elizabeth Mokwena, Edward Duncan

**Affiliations:** 1Department of Pharmaceutical Sciences, School of Pharmacy, Sefako Makgatho Health Sciences University, Pretoria 0204, South Africa; 2Department of Public Health Pharmacy and Management, School of Pharmacy, Sefako Makgatho Health Sciences University, Pretoria 0204, South Africa; hannelie.meyer@smu.ac.za; 3South African Vaccination and Immunisation Centre, Sefako Makgatho Health Sciences University, Pretoria 0204, South Africa; 4Department of Public Health, School of Healthcare Sciences, Sefako Makgatho Health Sciences University, Pretoria 0204, South Africa; kebogile.mokwena@smu.ac.za; 5Faculty of Health Sciences and Sport, University of Stirling, Stirling FK9 4LA, UK; edward.duncan@stir.ac.uk

**Keywords:** academic staff, burnout, Global South, mental health promotion, organisational culture, psychological distress, university employees, workplace mental health

## Abstract

Mental health challenges in academic settings are increasingly recognised, yet research on staff wellbeing remains limited, particularly within African universities. This study provides the first institution-wide assessment of psychological distress and burnout among academic staff at a South African university. A cross-sectional survey using validated tools, the 28-item General Health Questionnaire (GHQ-28 ) and the Oldenburg Burnout Inventory (OLBI) was administered to 157 academic employees, and data were analysed using descriptive statistics, non-parametric tests, and ordinal regression. The median age of participants was 42 years (*Interquartile range [IQR]* = 34–50; *SD* = 11.4), and the majority of participants were female (*n* = 110, 70%). The sample included staff across academic ranks, with lecturers being the most common (*n* = 64, 41%). Results showed that nearly half of participants (49%) exhibited severe psychological distress, and over a quarter (27%) reported high levels of burnout. Female staff reported significantly higher distress and burnout scores compared to their male counterparts. Less than a third (28%) of participants reported feeling safe to disclose mental health concerns, while over half expressed dissatisfaction with institutional support. Participants indicated strong support for both individual-level services, such as confidential counselling and workshops, and systemic changes, including flexible work arrangements and leadership-driven mental health initiatives. Findings highlight the need for integrated, participatory mental health strategies that are culturally and contextually tailored. These results offer timely evidence to inform the development of institutional strategies, policies, and practices to promote mental health among academic staff.

## 1. Introduction

The importance of wellbeing in the workplace is recognised by the World Health Organization (WHO), highlighting significant productivity concerns as well as societal and economic impacts associated with poor mental health ([82]). The cost of poor mental health to the global economy is estimated at USD 1 trillion lost annually in productivity. Thus, poor mental health should not be seen as an individual-level priority, but the macroeconomic burden should also be considered.

Key constructs underpinning this study are psychological distress, burnout and organisational culture. Psychological distress refers to an emotional state marked by symptoms such as anxiety, depressive symptoms and social dysfunction, often triggered by work pressures ([78]; [1]). Burnout is a chronic work-related syndrome of exhaustion, disengagement and reduced professional efficacy ([71]). Burnout has recently been reconceptualised with exhaustion, mental distance, and cognitive/emotional impairment as core dimensions. Organisational culture refers to the shared values and practices that shape how employees experience support for wellbeing and mental health ([17]; [40]) while the related construct of psychological safety refers to the belief that the work environment is safe for interpersonal risk-taking, and is conducive for employees to raise concerns and challenges without fear ([57]; [22]).

The Job Demands–Resources (JD-R) model provides a useful theoretical anchor: excessive job demands, coupled with limited organisational resources, heighten risks of stress and burnout. Organisational culture and psychological safety can therefore act as critical contextual resources, buffering or exacerbating strain.

Research in higher education globally shows that academics face high levels of stress, often balancing teaching, research, mentorship, and administrative duties ([44]; [10]). Women, in particular, experience disproportionate strain linked to gendered caregiving roles and work–family conflict. Studies consistently find higher rates of emotional exhaustion and burnout among female academics ([84]; [63]; [6]). Longitudinal research further shows that early-career women display steeper burnout trajectories than men, reflecting a “double burden” of professional and domestic responsibilities ([25]). These global patterns underscore gender as a persistent determinant of wellbeing in academia.

There is a growing body of evidence on mental health and wellbeing in higher education settings; however, most prior studies are situated in the Global North, often described as being conducted among faculty, typically aggregating academic roles. These studies frequently aggregate academic roles, emphasise burnout or stress rather than broader wellbeing, and rarely consider institutional culture ([44]; [63]). High rates of psychological distress and burnout are consistently reported ([34]; [35]) but often fail to provide contextual insights. Despite evidence that workplace norms influence wellbeing and comfort with disclosure ([35]; [48]), the role of an organisational culture remains underexplored.

This is especially true of African settings, where recent evidence is emerging. For example, burnout has been linked with productivity loss among Egyptian academics ([5]), and Ugandan research highlights the prevalence of burnout among private university teaching staff ([83]). In South Africa, a recent narrative review underscored urgent wellbeing challenges, including excessive workloads, inadequate resources, and workplace bullying ([19]). Together, these studies demonstrate that academic staff in African contexts are at considerable risk, but they often focus on specific aspects (such as burnout alone or wellbeing challenges in general) and rarely combine validated measures of distress and burnout with institutional culture and staff preferences for support.

While international and African studies highlight high distress, these dynamics must be situated within South Africa’s unique higher education context. South Africa’s higher education environment is shaped by historic, systemic and institutional factors that persist despite decades of reform. The legacy of apartheid continues to influence institutional cultures while transformation-based policies aimed at redressing social and racial inequalities have shown slow progress, leading to ongoing tensions and emotional strain for students and staff alike ([41]; [56]). Governance crises are common, especially in historically disadvantaged institutions (HDIs), compounded by structural challenges, such as high student–staff ratios and an ageing academic workforce ([21]). HDIs refer to universities that were established under apartheid to serve Black, Coloured, and Indian populations, and were systematically underfunded and marginalised compared to historically white institutions. These legacies continue to shape inequalities in infrastructure, staffing, research output, and institutional culture despite decades of reform ([3]; [9]).

South African universities are also seen as conducive spaces for debating elements of the transformation agenda such as social justice, equity and decolonisation of higher education; when these initiatives are slow to show tangible progress, these may also contribute to emotional fatigue and burnout amongst staff ([49]; [28]).

Student protests remain a defining feature of the South African higher education landscape, with demands ranging from opposition to high fees and unstable student enrolment to calls advocating for the decolonisation of higher education. Academic staff are often caught in between institutional resistance and social justice, adding to their psychological burden. In some HDIs, this strain has been exacerbated by reports of corruption and instability with negative effects on staff morale and retention ([61]; [76]; [41]). The study site itself also has a long history of student and staff protests, situating the current research within this challenging institutional context. Prior studies largely emphasised stress or burnout and overlooked institutional culture; our study extends this work by integrating validated measures with organisational culture and staff preferences in a South African HDI context.

### 1.1. Present Study and Contribution

To our knowledge, this is the first institution-wide survey of mental health, burnout and organisational culture among academic staff in a historically disadvantaged university in South Africa. We integrate validated measures of psychological distress (GHQ-28) and burnout (OLBI) with items on organisational culture and staff preferences for mental-health support—linking individual outcomes with institutional determinants. This combined approach extends prior work, which has tended to focus on single outcomes, narrow staff groups or review-level synthesis, and offers context-specific evidence for feasible and acceptable wellbeing responses at HDIs. In sum, earlier studies highlight distress and burnout among academics but provide limited insight into the role of organisational culture, especially within African universities. Gender and career-stage differences have been noted internationally but remain underexplored locally. To address these gaps, we tested four hypotheses relating distress, burnout, gender, experience, and organisational culture.

### 1.2. Aims and Hypotheses

The study aimed to (1) assess levels of psychological distress and burnout; (2) explore institutional culture regarding mental health; and (3) identify staff preferences for support. Guided by the Job Demands-Resources model ([16]) and prior evidence on gender disparities ([84]; [25]), early career vulnerability ([63]) and the role of organisational culture and psychological safety ([22]; [17]), we proposed the following hypotheses:

**H1.** 
*Higher psychological distress (GHQ-28) is positively associated with higher burnout (OLBI).*


**H2.** 
*Female staff report higher levels of distress and burnout compared to male staff.*


**H3.** 
*Staff with fewer years of experience report higher distress and burnout than longer-tenured staff.*


**H4.** 
*Negative perceptions of organisational culture are associated with higher distress and burnout.*


## 2. Materials and Methods

### 2.1. Study Setting and Design

This was a cross-sectional study design, using an electronic survey administered among academic staff at a South African University. The institution is historically disadvantaged and has approximately 6000 students enrolled across five Schools: Medicine, Pharmacy, Dentistry, Healthcare Sciences and Science and Technology. There are over 600 academic employees employed at the institution, with 200 permanently employed, while the remainder comprise either joint appointments with the National Department of Health, or contract academic appointments. The institution offers both undergraduate and postgraduate programmes, which are primarily focused on health science disciplines.

### 2.2. Target Population and Sample

Inclusion criteria for the study population were set as any permanently employed staff member at the university who was directly involved in teaching and/or supervising undergraduate and postgraduate students.

The minimum sample size calculation was performed using the online sample size calculator Raosoft (http://www.raosoft.com/samplesize.html, accessed on 11 November 2022). The total number of academic employees at the institution was approximately 619 (200 permanent and 419 joint appointments) based on the information obtained from the University Human Resources Department. We used the following parameters to calculate the sample: 95% confidence level, 7% margin of error and a response distribution of 50%. Thus, the minimal required sample size was calculated to be 150 participants.

### 2.3. Measures

The survey included five distinct sections, an introductory section covered socio-demographic (e.g., age, gender, etc.) and professional information (e.g., years of experience, highest qualifications, etc.). Section A1 included the 28-item General Health Questionnaire, a licensed screening tool for psychological distress (available from ePROVIDE by Mapi Research Trust: https://eprovide.mapi-trust.org/instruments/general-health-questionnaire, accessed on 15 February 2023), while Section A2 included the Oldenburg Burnout Inventory (OLBI), a validated instrument for assessing work-related burnout, used with permission from the tool authors and available online (https://www.carepatron.com/files/oldenburg-burnout-inventory.pdf, accessed on 15 February 2023). Appendix A were both developed de novo by the research team to measure organisational culture regarding mental health and identify mental health needs in this population, respectively. Items in this section were developed through team consultation, based on standard workplace mental health audits and tailored to the local academic context after iterative review and refinement. Items were reviewed by a panel of three experts who are the PhD supervisors, academic leaders and specialists in the field of public and mental health for clarity, relevance and alignment with the study’s purpose as well as institutional context. Iterative drafts of the questionnaire were distributed, leading to minor modifications such as simplifying wording, reordering questions and the addition of open-ended items to achieve a better spread of responses. Although content validity was not formally assessed, the process provided face and content validity evidence.

The survey also underwent cognitive pre-testing amongst a small subset (n = 10) of part-time junior lecturers and lecturers from the School of Pharmacy. Feedback from this group further allowed for refinement of the wording and flow of the survey items and confirmed item clarity. Data from the pilot test (n = 10) were excluded from the final analysis to ensure only full survey responses were included. Due to copyright and licensing restrictions, only the sections created de novo (Appendix A) are included as Appendix A; the GHQ-28 and OLBI were used under license/through author permissions and thus are not reproduced here.

#### 2.3.1. The General Health Questionnaire-28 (GHQ-28)

The General Health Questionnaire-28 was developed by Goldberg and Hillier as a screening tool used for detecting non-psychotic psychiatric disorders ([31]). The 28-item tool is designed to assess four aspects of psychological distress, namely, somatic symptoms, anxiety/insomnia, social dysfunction and depression (7 items each) ([32]; [39]). The questionnaire is structured using a 4-point Likert scale with options such as “Not at all”, “No more than usual”, “Rather more than usual” and “Much more than usual”. Examples items include “*Have you recently been feeling perfectly well and in good health?*” (somatic symptoms), “*Have you recently lost much sleep over worry*” (Anxiety/insomnia), “*Have you recently felt constantly under strain*” (social dysfunction) and “*Have you recently felt that life is entirely hopeless?*” (depression).

The GHQ-28 has been successfully translated to a variety of languages, contexts and occupations ([52]; [39]; [14]; [36]). and there are also examples of its use and applicability in the academic context ([80]; [38]; [43]). The unmodified English version of the survey was used in this study, as English is the medium of instruction at the university.

The GHQ-28 was scored using Binary scoring (0-0-1-1) instead of the Likert scoring (0-1-2-3) in line with recommendations from the literature ([27]; [2]). Higher scores are associated with higher levels of psychological distress; the total score range is between 0 and 28 when using the binary scoring (0-0-1-1) method. While a threshold of >5 is widely used to identify probable cases of distress in general populations, validation studies of the GHQ-28 across diverse cultural and occupational groups have shown that optimal cut-offs vary by context ([36]; [53]; [54]). Because our study focused on a non-clinical, occupational sample, we stratified total scores into mild (0–4), moderate (5–7), and severe (8–28) categories to capture the continuum of functional strain in the academic workforce. This approach is consistent with evidence that GHQ severity bands should be tailored to population and setting ([69]; [26]), and was intended to provide more granular insight into staff wellbeing rather than to establish diagnostic thresholds. These categories should therefore be interpreted as context-specific indicators of strain rather than clinical diagnoses.

Internal consistency in this study was excellent (Cronbach’s α = 0.97, 95% *CI*: 0.95–0.98). Subscale reliabilities were strong for Social Dysfunction (*α* = 0.86) and Depression (*α* = 0.82). For Somatic and Anxiety/Insomnia, item variability was low and *α* could not be reliably estimated. A confirmatory Factor Analysis was not conducted; therefore, model fit indices (e.g., common fit index, Tucker-Lewis Index, Root Mean Square Error of Approximation, Standardized Root Mean Square ResidualI could not be reported.

A licence and permission to use the GHQ-28 in this study were obtained from Mapi Research Trust, Lyon, France. (https://eprovide.mapi-trust.org, accessed 15 February 2023).

#### 2.3.2. The Oldenburg Burnout Inventory

The Oldenburg Burnout Inventory (OLBI) was developed by Demerouti, Bakker, Nachreiner, and Schaufeli as an alternative to the Maslach Burnout Inventory in order to better capture burnout across broader occupations ([16]). The tool consists of 16 items, 8 measuring disengagement and 8 for exhaustion ([33]). Scoring of items involves a 4-point Likert scale (ranging from strongly agree to strongly disagree), and higher scores indicate higher levels of burnout ([67]). Examples of items under the two domains: “*There are days when I feel tired before I arrive at work*” (Exhaustion domain) and “*I always find new and interesting aspects in my work*” (disengagement domain). Negatively worded items have been included to reduce response bias.

The OLBI has shown good internal consistency across multiple studies and populations and has proven to be a valid tool ([74]; [72]; [66]; [60]). It has also been used effectively in academic settings ([51]). In this study, we used the unmodified English version as provided by the tool authors. Internal consistency in this study was excellent (α = 0.91, 95% *CI*: 0.82 to 0.96). Subscale-level alphas for OLBI were less stable, consistent with ongoing debates about OLI dimensionality and mixed item wording; therefore, we emphasise the overall reliability estimate. Confirmatory factor analysis was not conducted for this dataset, and model fit indices could not be reported, which is acknowledged as a limitation of the study.

Permission to use the OLBI was obtained from the tool author via email correspondence; evidence of this can be made available upon request.

#### 2.3.3. Institutional Culture Regarding Mental Health

The perceptions of academic staff regarding the institutional culture around mental health were assessed by questions based on standard workplace mental health audits ([81]; [11]) and adapted to the local context after expert review, as outlined in Section 2.3, to ensure contextual relevance and clarity. This section consisted of 9 Likert scale of agreement questions (1 = Strongly Disagree to 5 = Strongly Agree) on topics such as their comfort disclosing mental health information at work, and perceptions of the institution’s actions, policies and overall climate regarding mental health support and services. As such, this section was not designed with a scoring system in mind, nor to calculate a composite score, but data were analysed to identify patterns and preferences of participants in order to map systemic facilitators and/barriers to better inform the development of the intervention design that is part of the broader PhD project. An open-ended optional question was included at the end of this section for participants to provide additional comments on the institutional culture and its impact on academic staff mental health.

#### 2.3.4. Needs Assessment for Workplace Health and Wellbeing

This section assessed staff preferences regarding potential mental health promotion strategies at the institution. Eight questions were included in this section. Four Likert-scale of agreement questions were used to assess staff support for mental health initiatives (peer-based support, technology-based resources, organisational culture and external partnerships) with answer options ranging from 1 = Strongly Disagree to 5 = Strongly Agree. Three questions allowed for multiple responses on preferences for mental health resources and broader changes that could promote positive mental health at the institution. Participants were also given the option to add their own suggestions to these, and finally, one open-ended question was included to capture any additional suggestions or recommendations related to the mental health needs of academic staff. As described in Section 2.3, item development followed expert review and cognitive pre-testing to establish face and content validity. Similar to Appendix A, this section was not designed to produce a summary score, but to identify staff preferences for intervention types and offer respondents the opportunity to suggest possible interventions. The findings of Appendix A also served to guide and define topics for inclusion in a later qualitative phase of the PhD project, which falls outside of the scope of the current paper.

### 2.4. Recruitment Strategy, Survey Administration and Ethical Considerations

Data collection was performed between January and May 2024 using a self-administered questionnaire distributed via email. Information about the study was disseminated to all Deans of Schools and Directors of Units in the university for further dissemination within Departments. After this initial dissemination, there was a very slow uptake in participation, and in order to ensure that a representative sample would be achieved, further recruitment was deemed necessary. To this end, additional recruitment was performed through presentations about the study at School Board Meetings and face-to-face recruitment was also performed at a two-day Wellness Day event for staff and further to this, as well as door-to-door canvassing. Although recruitment strategies were varied (virtual and face-to-face), all responses were submitted by participants using the online questionnaire.

The email invitation for participants contained a link to the survey held on OnlineSurveys, as well as a participant information sheet to fully inform potential participants about the purpose of the study. Access to OnlineSurveys was provided by the University of Stirling. The information sheet and front page of the online survey explicitly informed potential participants that the survey was confidential and that participation was voluntary. Participants were also informed that information collected would be protected by the General Data Protection Regulation (GDPR) in the United Kingdom and the Protection of Personal Information Act (POPIA) in South Africa, as this study is part of a PhD project that is registered both in South Africa and in Scotland. Participants were also informed that they could withdraw from the study at any time by leaving the survey before submission and that the platform OnlineSurveys would not collect their responses. Acceptance of participation on the first page of the survey was considered as informed consent. The information leaflet and front page of the survey outlined the minimal risks of participating in the survey (potential discomfort with subject matter). Contact details of support resources were provided at the beginning and end of the survey. Ethical approval for this study was obtained from the Sefako Makgatho University Research Ethics Committee (SMUREC/P/116/2022: PG) and the University of Stirling General University Ethics Panel (NICR 2023 10971 10174). All procedures complied with the Declaration of Helsinki and applicable data protection regulations (GDPR and POPIA).

### 2.5. Data Management and Statistical Analysis

Analyses were conducted to test the hypotheses outlined in the introduction. Survey responses were downloaded from the OnlineSurveys platform in Microsoft Excel format. (Excel for Microsoft 365). As the survey tool was designed to prevent item non-response, all submitted responses were complete, and no data imputation or cleaning was required. Demographic characteristics and item responses were summarised using descriptive statistics. Medians and interquartile ranges (*IQR*) were reported for continuous variables (e.g., age, questionnaire scores), while categorical variables (e.g., gender, rank, qualification) were presented as frequencies and percentages.

Since the GHQ-28 and OLBI scores both had non-normal distributions, non-parametric tests were used to examine potential associations between the GHQ-28 and OLBI scores and demographic variables. Quantile linear regression was used to assess associations between continuous variables and the conditional median of the outcome variables. Wilcoxon-rank sum tests were used for comparisons across categories (e.g., gender) and where more than two groups were compared (e.g., designation, qualifications) Kruskal–Wallis tests were applied.

The relationship between GHQ-28 scores (levels of psychological distress) or OLBI scores (levels of burnout) and levels of agreement with statements related to institutional culture was analysed using separate ordinal logistic regressions for each score. A *p*-value of <0.05 was considered statistically significant. All analyses were conducted using R software (version 4.4.0).

As mentioned previously, open-ended responses at the end of Appendix A were not subjected to formal analysis for this paper. These questions were included in the survey to elicit formative insights to guide the subsequent qualitative phase of the broader doctoral study and to inform the development of the future mental health promotion intervention in this setting. Thus, the responses to these questions are not reported in this manuscript.

## 3. Results

### 3.1. Participant Characteristics

A total of 157 participants completed the survey (N = 157). The median age was 42 years (*IQR* = 34–50). Most participants identified as female (n = 110, 70.1%), followed by male (n = 44, 28.0%). The most common academic rank was lecturer (n = 65, 41.4%) with representation from all Schools, most commonly the School of Healthcare Sciences (n = 46, 29.3%). Just over a quarter of participants (n = 42, 26.8%) had ≤5 years of academic experience. A full summary of participant demographic characteristics is presented in Table 1.

### 3.2. Psychological Distress (GHQ-28 Scores)

The following severity thresholds applied: mild distress (score between 0 and 4), moderate distress (score between 5 and 7) and severe distress (score above 8). Total GHQ scores ranged between 0 and 27, with *M* = 8.59 (*SD* = 6.07), *Mdn* = 7 (*IQR*: 1–16), which indicates moderate distress on average. Notably, nearly half of the respondents’ (n = 77, 49.0%) scores were suggestive of severe psychological distress. Just over a third of respondents (n = 59, 37.1%) scored in the category of mild distress, while a small proportion (n = 21, 13.2%) scored in the category of moderate distress. The distribution of scores was right-skewed, indicating that while many staff reported low or moderate distress, a substantial subgroup experienced very high scores.

### 3.3. Burnout (OLBI)

The OLBI was scored using a 1–4 Likert scale, and reverse scoring was applied to eight items in line with the author’s scoring guidelines ([15]) and literature ([72]; [67]). Total OLBI scores ranged from 0 to 61, *M* = 39.75 (*SD* = 9.17), *Mdn* = 39 (*IQR*: 35–44), indicating moderate burnout on average.

Based on published thresholds ([30]), where OLBI scores <30 indicate low burnout, 30–44 moderate burnout, and >44 high burnout, the distribution in this sample suggests that most respondents experienced moderate to high levels of burnout. Nearly a third of respondents screened in the high burnout category (n = 43, 27.3%), while the majority of respondents fell within the moderate burnout range (n = 98, 62.4%). Only a small proportion of respondents (n = 18, 11.5%) were classified as experiencing low burnout. The elevated median score and distribution pattern indicate that burnout was widely experienced across staff rather than confined to a small subgroup.

### 3.4. Associations Between GHQ-28 Scores, OLBI Scores and Demographic Variables

Table 2 summarises the associations between demographic variables and psychological distress (GHQ-28) and burnout (OLBI) scores. A statistically significant association was observed only for gender. Female participants had higher GHQ-28 scores (*Mdn* = 9.5, *IQR* = 3–18) compared to males (*Mdn* = 3.5, *IQR* = 1–10; *p* = 0.004). Similarly, OLBI scores were higher among females (*Mdn* = 41, *IQR* = 36–45) than males (*Mdn* = 37, *IQR* = 33–40; Wilcoxon *p* = 0.002). All other demographic variables, including age, rank, qualifications, years of academic experience, and School, showed no statistically significant associations with either measure.

### 3.5. Institutional Culture Regarding Mental Health

Overall, participants reported concerns and dissatisfaction regarding psychological safety, transparency, and institutional commitment to staff wellbeing. Less than a third of participants (n = 44, 28.0%) agreed or strongly agreed that they felt comfortable disclosing information regarding mental health at work, while more than half (n = 83, 52.9%) disagreed or strongly disagreed. There was a high level of disagreement amongst participants regarding the following: institutional provision of sufficient training and support for mental wellbeing (n = 105, 66.9%); adequate mental health services (n = 104, 66.2%) or whether the institution has an established culture of caring for staff (n = 108, 68.8%). Less than a fifth of participants (n = 17, 10.8%) agreed that policies and procedures were established in the institution, while the majority (n = 65, 41.4%) remained neutral in their response to this dimension. Table 3 provides a detailed breakdown of responses to these items.

Ordinal regression models (adjusted for age, gender, and academic rank) confirmed that higher psychological distress (GHQ-28) and higher burnout (OLBI) were significantly associated with lower odds of agreement across all five institutional culture items (Q48–Q52). GHQ-28 odds ratios ranged from 0.86 to 0.94 (all *p* ≤ 0.02), and OLBI odds ratios ranged from 0.89 to 0.94 (all *p* < 0.001), indicating that staff with greater distress and burnout were less likely to perceive the university as supportive in terms of disclosure, training and services, culture of care, and policy development (Table 4). Additional institutional items (Q53–Q56) showed a similar pattern, with both GHQ-28 and OLBI scores significantly predicting lower odds of agreement across equity, empowerment to seek support, positive working practices, and work–life balance (all *p* < 0.001).

### 3.6. Preferences for Institutional Interventions

Participants were asked to indicate their preferences regarding types of mental health support they would like to see implemented at the institution. Overall responses indicated strong support for both individual and organisational level interventions. The most frequently selected options were confidential counselling services (n = 130, 82.8%), workshops on stress and resilience (n = 122, 77.7%), mental health awareness campaigns (n = 96, 61.1%), and peer support groups (n = 82, 52.2%). Additional suggestions included proactive strategies, mentorship, team-building activities, and a dedicated wellness centre, with an emphasis on flexible delivery formats (e.g., both physical and online options). There were no statistically significant differences in GHQ-28 or OLBI scores between those who selected and those who did not select any of the listed services (all *p* > 0.05), suggesting that preferences were broadly shared across levels of distress and burnout. Comprehensive details around staff preferences are presented in Table 5.

## 4. Discussion

These findings confirm and extend the theoretical expectations outlined earlier, particularly the JD–R model and gendered patterns of academic strain. The results of this study provide a local benchmark where psychological distress and burnout are high in our sample. This is the first study focusing on the mental health of academics in this institution, which is nearly 50 years old. Rates of psychological distress in our sample exceed those published in recent international studies, where around a third of academics experienced high levels of distress ([68]; [34]). These findings emphasise the need for targeted context-specific support to address the significant burden of poor mental health in higher education settings.

Burnout levels in this study were also elevated compared to reports from some high-income contexts ([64]; [8]) but were more comparable to recent findings from Brazil ([4]). These similarities may reflect parallel challenges in resource allocation between South Africa and Brazil, both of which face structural constraints in higher education systems. A unique contextual factor in this setting is the prevalence of dual academic–clinical roles, whereby staff are simultaneously responsible for teaching, research, and clinical service delivery. These overlapping responsibilities substantially intensify workload demands, leaving less time for recovery and contributing to the elevated levels of burnout observed in this study. Such role multiplicity has been shown to amplify emotional exhaustion and reduce work–life balance, as clinical obligations are often unpredictable and high-stakes, while academic duties require sustained cognitive and administrative engagement. The intersection of these domains may therefore create a “compounding strain,” where pressures from one role spill over and exacerbate challenges in the other. Health sciences universities are particularly affected because clinical service is embedded into institutional expectations, and staff shortages further heighten the burden. Evidence from recent studies confirms that academics who combine teaching, research, and clinical responsibilities report significantly higher burnout and poorer wellbeing outcomes compared to their non-clinical peers ([77]; [45]; [59]). These findings situate the current results within the specific organisational ecology of a health sciences university, where dual role demands constitute a critical driver of psychological strain.

Gender emerged as the only significant predictor of both psychological distress and burnout in this study. This finding is consistent with international evidence showing that women in academia face disproportionate risks of poor mental health outcomes, including greater emotional exhaustion and lower work–life satisfaction ([79]; [12]; [25]). These disparities are rooted in longstanding gender norms that place disproportionate caregiving and domestic responsibilities on women, which produce elevated levels of work–family conflict ([84]). Work–family conflict is now recognised as a central predictor of burnout, with women consistently reporting higher levels than men and those in the high WFC group experiencing burnout rates 25–30% greater than their peers ([84]). The COVID-19 pandemic further magnified these challenges, as female academics reported increased emotional labour and heightened vulnerability to burnout ([20]).

An intersectional lens helps to explain how these gendered burdens are compounded by additional structural inequities, particularly in resource-constrained or historically disadvantaged institutions, where women may also contend with barriers to career progression, precarious contracts, or racialised exclusion. South African evidence reinforces these patterns, with female academics reporting heightened burnout ([7]) and frequent perceptions of inadequate institutional support ([58]). Importantly, women who perceived their institutional climate as equitable reported substantially lower burnout than those who did not ([18]), underscoring the role of organisational culture in either amplifying or buffering gendered disparities.

Our findings, therefore, underscore the importance of embedding gender-sensitive and equity-focused strategies within institutional policy frameworks rather than relying on isolated wellness initiatives. In the South African context, this includes ensuring equitable workload distribution, supporting caregiving responsibilities, and addressing barriers to career advancement that disproportionately affect women. Situating these strategies within broader theoretical perspectives on work–family conflict and intersectionality highlights that gender disparities in academic wellbeing are not simply individual vulnerabilities, but the result of structural, cultural, and institutional dynamics that require systemic solutions.

In our study, less than a third of participants felt safe disclosing mental health concerns, and higher GHQ-28 and OLBI scores predicted lower comfort levels. This reflects a systemic lack of psychological safety in the institution. Psychological safety, the perception that one can speak up without fear of reprisal ([22]), is not just a background feature of organisational climate but a causal mechanism linking culture to mental health. When staff anticipate stigma or judgement, they internalise silence, avoid help-seeking, and disengage from institutional life, reinforcing cycles of stress and burnout ([29]; [73]).

Stigma compounds this dynamic. Disclosure of mental health concerns in academia is often framed as a personal weakness, which undermines credibility and career prospects ([13]; [75]). In contexts of low psychological safety, stigma operates as an institutional norm that constrains behaviour, not just an individual attitude. Our data illustrate this interaction: those with the highest levels of distress and burnout also reported the lowest comfort disclosing, showing how stigma and psychological safety together operate as institutional mechanisms that translate high job demands into poorer mental health outcomes.

Leadership and organisational climate play a decisive role in either breaking or reinforcing these cycles. Leaders who model vulnerability, encourage open dialogue, and normalise help-seeking can counter stigma and create psychologically safe environments ([57]; [37]). In contrast, when leadership visibility is low or policies are invisible, staff interpret this as evidence that mental health is not valued, further deepening mistrust. This may explain why, even where wellness policies exist in our setting, participants described them as absent or ineffective.

Thus, the interplay of psychological safety and stigma in this study highlights how institutional culture actively shapes health outcomes. Addressing poor mental health in academia requires not only individual-level services but also structural interventions that reduce stigma and embed psychological safety into everyday practices, such as transparent communication, feedback mechanisms, and participatory policy design.

Stigma further reinforces the dynamics described above, operating as a system-level barrier that prevents disclosure and service use. In our data, this was reflected in low agreement with statements about institutional care and psychological safety. Participants were reluctant to disclose mental health concerns due to anticipated consequences or judgement, consistent with findings from similar academic institutions ([73]). Stigma in this context should not be understood solely as individual prejudice, but rather as a cultural norm embedded within academic environments that suppresses help-seeking and normalises silence ([13]; [75]).

Such stigma is reinforced by the competitive and high-pressure culture of academia, where performance demands and perfectionist norms leave little room for open discussion of wellbeing ([24]). Early career researchers may be particularly vulnerable, fearing reputational damage and stalled career progression if they disclose difficulties ([47]). Our findings are consistent with this pattern, showing that younger academics reported elevated levels of distress and burnout. Within the JD-R framework, this reflects how excessive demands, coupled with limited institutional resources and support, can disproportionately strain early career staff.

Another salient finding was the invisibility, or lack of awareness of institutional policies addressing mental health. Many participants were unaware of existing wellness policies, reflecting weak communication and limited leadership visibility. Poorly disseminated policies risk creating confusion, fragmented responsibility, and ultimately poorer outcomes for staff ([70]). In our context, policy invisibility also reflects a lack of institutional support and buy-in, which compounds low psychological safety and entrenched stigma. These dynamics highlight that policies alone are insufficient unless they are clearly communicated, enacted, and embedded in institutional culture, particularly in historically disadvantaged universities where trust in leadership may already be fragile.

Although a formal employee wellness policy exists at the study site, the lack of staff awareness suggests weak dissemination and implementation. Without clear rollout plans, even well-designed policies often fail to influence institutional culture ([62]). In practice, a policy that is present but invisible can signal tokenism and inadvertently convey that staff wellbeing is not a genuine institutional priority. Within the JD-R framework, such invisibility represents a loss of critical organisational resources, leaving high demands unbuffered and further contributing to burnout and psychological distress.

A notable finding was that nearly 70% of participants in this study endorsed the use of regular feedback mechanisms to improve mental health services. This reflects growing demand for inclusive and participatory approaches to policy development—approaches that are transparent, responsive, and co-created with staff. There are similar recommendations that institutional mental health policies be developed through consultation and made accessible through clear, ongoing communication, in order to foster trust and uptake ([65]). Such participatory approaches may also help rebuild trust in leadership and foster psychological safety.

Participants expressed wide-ranging preferences for workplace-based mental health services, spanning both individual-level support and systemic reforms. Confidential counselling and workshops were the most endorsed services, echoing global findings that employees prioritise confidential and skill-building initiatives ([55]). Participants also emphasised the importance of co-design, reinforcing the need for initiatives that include employee input from the outset. At the same time, participants called for peer support groups, flexible work and leadership support. Leadership visibility and modelling are critical for uptake and credibility ([42]), while employee participation in policy co-creation has been emphasised; these efforts also require top-down endorsement for successful engagement ([65]). These diverse perspectives highlight the importance of integrated approaches that span across individual, team and organisational levels ([46]). They also reflect the structural challenges of low- and middle-income country (LMIC) settings, where inequalities, workload pressures, and fragmented services demand tailored and sustainable solutions ([23]; [50]).

Taken together, our findings indicate a misalignment between the mental health needs of academic staff and the support structures currently available within the institution. While participants expressed strong preferences for both individual- and system-level interventions, the lack of psychological safety, visible leadership engagement, and policy clarity suggests a need for strategic cultural change. Beyond isolated services, there is a need in this specific institution to cultivate a safe, inclusive and transparent environment, one where mental health is embedded into everyday practices, not treated as an ancillary concern. Our findings contribute by demonstrating how validated measures of distress and burnout can be meaningfully linked with institutional culture and staff preferences to identify such points of misalignment.

There are some limitations to consider when interpreting the findings of our study. Firstly, the research was conducted at a single institution, which limits the generalisability of the results to other academic or geographic contexts. However, the findings offer deep, context-specific insights that may resonate with other under-resourced higher education institutions, particularly within LMICs. Secondly, self-reported data formed the basis of this study, which has the potential to introduce response bias, including social desirability or underreporting of mental health concerns. To mitigate this, validated instruments were used alongside anonymous data collection to encourage openness and accuracy. The high levels of psychological distress and burnout reported further suggest that participants were willing to disclose their experiences candidly, reducing concerns of underreporting. Thirdly, although the items developed de novo to assess institutional culture and needs assessment were reviewed by experts and pilot-tested for clarity, content validity was not formally assessed, and this should be considered when interpreting findings from these sections. Finally, the cross-sectional design limits causal inference and confirmatory factor analysis was not conducted for GHQ-28 and OLBI, which should be considered when interpreting psychometric robustness. Both scales showed excellent overall internal consistency, although subscale reliabilities were mixed. Future studies should formally test factor structures, report model fit indices and examine subscale reliability under alternative scoring approaches to strengthen psychometric evidence in South African academic populations. Nonetheless, the strength of associations observed and the alignment with existing literature reinforce the relevance and urgency of the findings.

## 5. Conclusions

There are high levels of psychological distress and burnout amongst academic staff at this South African university. The study identified limited psychological safety, low policy visibility, and inconsistent leadership engagement as institutional factors associated with poorer wellbeing. This is, to our knowledge, the first study to combine validated measures of distress and burnout with staff perceptions of organisational culture and intervention preferences within an African historically disadvantaged institution, offering novel and contextually grounded insights. These findings not only document the scale of the challenge but also highlight important pathways for theoretical and practical reflection.

### 5.1. Theoretical Implications

This study contributes by being the first institution-wide survey in a South African HDI to integrate validated measures of distress and burnout with organisational culture and staff preferences for support. Our findings extend the Job Demands–Resources (JD-R) model by showing how excessive demands (e.g., dual academic–clinical roles, high workloads, gendered caregiving expectations) coupled with insufficient resources (e.g., low psychological safety, invisible wellness policies) contribute to poor mental health outcomes. We also highlight the role of organisational culture as a contextual resource that can either buffer or exacerbate strain. By demonstrating these dynamics within a historically disadvantaged university in the Global South, this study adds a novel, context-specific perspective to existing models that foreground institutional culture alongside individual demands and resources. This conceptual framing also informs the practical steps that institutions can take.

### 5.2. Practical Implications

Addressing academic mental health requires more than individual services; it demands systemic and cultural change. Institutions should address the following:Improve the visibility, communication, and implementation of existing wellness policies.Strengthen psychological safety through leadership modelling, open communication, and stigma reduction.Embed gender-sensitive strategies into policy and workload allocation to address disproportionate burdens on women.Co-design mental health initiatives with staff to enhance trust, uptake, and sustainability.

In settings such as ours, feasible measures must come at a low cost. Potential strategies include training peer supporters, embedding wellness check-ins into routine meetings and increasing visibility of policies using digital channels. Flexible workload adjustments during peak periods can further reduce strain without major investment. Recent evidence suggests that such tailored and scalable approaches are more sustainable in LMIC settings than resource-intensive models ([23]; [50]). Finally, sustained leadership commitment and integrated action spanning across all levels are crucial to creating safer and more supportive academic environments.

## Figures and Tables

**Table 1 behavsci-15-01410-t001:** Participant Demographic Characteristics (N = 157).

Characteristic	*n (%)* or *Mdn (IQR)*
Age	42 (34–50)
Gender	
Female	110 (70.1%)
Male	44 (28.0%)
Non-binary	1 (0.6%)
Prefer not to disclose	1 (0.6%)
Transgender	1 (0.6%)
Highest qualification	
Diploma	6 (3.8%)
Bachelor’s Degree	12 (7.6%)
Honour’s Degree	8 (5.1%)
Master’s Degree	79 (50.3%)
PhD	52 (33.1%)
Years of experience in academia	
≤5	42 (26.8%)
6–10	39 (24.8%)
11–15	32 (20.4%)
16–20	17 (10.8%)
21–30	18 (11.5%)
≥31	9 (5.7%)
School in which employed	
School of Health Care Sciences	46 (29.3%)
School of Pharmacy	37 (23.6%)
School of Medicine	36 (22.9%)
School of Dentistry	15 (9.6%)
School of Science and Technology	15 (9.6%)
Other ^1^	6 (3.8%)
No response	2 (1.3%)
Job title	
Other ^2^	24 (15.3%)
nGap Lecturer	11 (7.0%)
Junior Lecturer	19 (12.1%)
Lecturer	65 (41.4%)
Senior Lecturer	26 (16.6%)
Associate Professor	5 (3.2%)
Professor (Full)	6 (3.8%)
Professor Emeritus	1 (0.6%)

*Note*. *Mdn* = median; *IQR* = interquartile range; *n* = sample size. ^1^ CUTL, Microbiology, Student Affairs. ^2^ Adjunct Professor, Clinical Training Internship Coordinator, Instructional Designer, Part-time Lecturer, Post-doctoral Fellow, Research Fellow, Senior Officer: Health/HIV, Senior Scientist, Staff Support, Tutor.

**Table 2 behavsci-15-01410-t002:** Associations Between Demographic Variables and GHQ-28 And OLBI Scores.

Demographic Variable	GHQ-28 *Mdn* (*IQR*)	OLBI *Mdn* (*IQR*)	Test	*p*-Value (GHQ-28/OLBI)
Gender (F vs. M)	9.5 (F) vs. 3.5 (*M*)	41 (F) vs. 37 (*M*)	Wilcoxon	** *0.004/0.002* **
Age (continuous)	-	-	Quantile regression	0.76/0.33
Academic rank	1–10	36–41	Kruskal–Wallis	0.78/0.81
Qualifications	*Dip* (2.0)—PhD (9.5)	35.5 (*Hons*)—41 (PhD)	Kruskal–Wallis	0.68/0.15
School	4–8.5	36–41	Kruskal–Wallis	0.91/0.42

*Note. Mdn* = median; *IQR* = interquartile range; *F* = female; *M* = Male; *Dip* = diploma; *Hons* = Honour’s degree. Bolded *p*-values indicate significance at *p* < 0.05.

**Table 3 behavsci-15-01410-t003:** Institutional Culture Responses (Q48–Q52).

Item	Disagree (%)	Neutral (%)	Agree (%)
Comfortable disclosing mental health	83 (52.9)	30(19.1)	44 (28.0)
The university provides training/support	105 (66.9)	36 (22.9)	16 (10.2)
The university provides adequate services	104 (66.2)	37 (23.6)	16 (10.2)
The university has a culture of caring	108 (68.8)	30 (19.1)	18 (11.5)
The university has developed policies/procedures	14 (8.9)	65 (41.4)	78 (49.7)

*Note.* Percentages reflect combined categories (Strongly disagree + Disagree; Agree + Strongly agree).

**Table 4 behavsci-15-01410-t004:** Ordinal Logistic Regression Results for Institutional Culture Items (Q48–Q56).

Institutional Culture Item	*OR* (GHQ)	95% *CI* (GHQ)	*p* (GHQ)	*OR* (OLBI)	95% *CI* (OLBI)	*p* (OLBI)
Q48. I feel comfortable disclosing mental health information within the university environment	0.94	0.89–0.99	0.18	0.94	0.91–0.98	0.001
Q49. The university provides training/support to enhance staff mental wellbeing	0.88	0.83–0.93	<0.001	0.89	0.86–0.93	<0.001
Q50. The university provides adequate mental health services	0.89	0.84–0.94	<0.001	0.89	0.86–0.93	<0.001
Q51. The university has a culture of caring that positively impacts mental health	0.86	0.81–0.92	<0.001	0.89	0.85–0.93	<0.001
Q52. The university has developed policies and procedures to address mental health concerns	0.88	0.83–0.93	<0.001	0.91	0.88–0.95	<0.001
Q53. The university actively promotes equity, diversity, and inclusion in mental health services	0.90	0.85–0.96	<0.001	0.92	0.89–0.95	<0.001
Q54. The university empowers academic staff to seek mental health support	0.90	0.85–0.96	<0.001	0.92	0.89–0.96	<0.001
Q55. The university creates positive working practices and conditions	0.88	0.83–0.93	<0.001	0.90	0.87–0.94	<0.001
Q56. The university promotes a healthy work/life balance for academic staff	0.88	0.84–0.94	<0.001	0.92	0.89–0.96	<0.001

*Note. OR* = Odds Ratio; *CI* = Confidence Interval. Models adjusted for age, gender, and academic rank. Lower *ORs* indicate that higher GHQ-28 and OLBI scores were associated with reduced odds of agreement.

**Table 5 behavsci-15-01410-t005:** Preferences for Institutional Mental Health Interventions (N = 157).

Preference Option	*n (%)*
Confidential counselling services	130 (82.8)
Workshops on stress and resilience	122 (77.7)
Mental health awareness campaigns	96 (61.1)
Peer support groups	82 (52.2)
Regular feedback mechanisms	109 (69.0)
Cultural sensitivity for providers	105 (66.5)
Tailoring of services	102 (64.6)
Multilingual resources	72 (45.9)
Flexible working hours	121 (76.6)
Institutional policies on work/life balance	117 (74.1)
Clear workload communication	106 (67.1)
Telecommuting options	101 (63.9)
Recognition of achievements	98 (62.0)

*Note.* Values are frequencies and percentages; multiple responses were allowed.

## Data Availability

The data that support the findings of this study are not publicly available due to institutional and ethical restrictions related to participant confidentiality. Anonymized data may be made available upon reasonable request and subject to approval by the relevant institutional ethics committees.

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
