# Peer review of "Workplace Mental Health Status Among Academic Staff: Psychological Distress, Burnout, and Organisational Culture at a South African University"

_behavsci, 2025, doi:10.3390/bs15101410_

Round 1

Reviewer 1 Report

Comments and Suggestions for Authors

The article presents a solid contextualization, with relevant references on mental health in university settings, especially in low- and middle-income countries. The theoretical background is extensive, although in some passages excessively detailed. A clearer synthesis of the central theoretical models is recommended to improve the flow of reading.

The methodological design is well outlined, with an adequate description of the instruments used (GHQ-28 and OLBI), the sample, and the statistical procedures. However, the hypotheses are not explicitly formulated. It is suggested that the authors present them in a clear and structured manner, in accordance with good empirical research practices.

The discussion is coherent and well articulated, with relevant links between the findings and the literature. There is balance in the interpretation of the data, although some passages repeat information already presented. Greater conciseness and focus on the practical implications of the results are recommended.

The findings are presented in a clear and objective manner, with appropriate use of tables and statistical analyses. Some tables could be accompanied by more direct interpretations in the text, facilitating understanding by non-specialist readers.

The article is well referenced, with relevant and varied sources. However, some references are dated (prior to 2010). The inclusion of more recent literature is suggested, especially on burnout and institutional mental health in Africa.

The conclusions are in line with the findings presented and are supported by the literature. It would be useful to distinguish more clearly between empirical conclusions and practical recommendations to avoid overlap between evidence and opinion.

The writing is clear and appropriate for academic style. A final linguistic review is recommended to improve fluency, especially in long and complex sentences.
In summary, the article has scientific merit and relevance, but requires specific adjustments in structure, methodological clarity, and bibliographic updates.

Reviewer 2 Report

Comments and Suggestions for Authors

Dear, researchers

In the introduction, concepts such as psychological distress, burnout, mental health, and organizational culture, which are central to the study, were not explicitly defined. There is also a lack of explanation as to why these variables should be explored. In addition, even though the subjects of the study were South African faculty, the job characteristics or institutional context of the group were not sufficiently revealed in the introduction. The contextual persuasive power of the study will be further strengthened by presenting a background explanation of the unique working environment or constraints that professors face.
In addition, since gender differences were confirmed in the results of this study, briefly presenting previous studies on the psychological burden between male and female professors and the gap in burnout experiences in the introduction will help increase logical connectivity. Finally, there is a lack of review of previous studies conducted on professors and an explanation of how this study differs from them. Beyond the description of "lack of professors' mental health research," the originality and contribution of the study will become clearer if it clearly summarizes the scope and limitations covered by the existing research and suggests how this study supplements them.

The basis of the hypothesis and the establishment of specific hypotheses have not been established in this study; therefore, the survey system appears somewhat insufficient. In particular, as we verify the differences between groups and the relationship between variables through statistical analysis, it is necessary to present a clear hypothesis, not just to outline research goals. In this regard, it is desirable to supplement the relationship between each variable in detail based on the review of previous studies in the theoretical background and to derive a hypothesis based on this.

In the discussion section, comparisons and interpretations with several previous studies are made. Still, the overall context is not smoothly continued due to an insufficient review of related studies in the theoretical background, and the academic contribution is not sufficiently revealed, as it is presented somewhat abruptly. In addition, it needs to be supplemented that clear theoretical implications for which part or new results this study has been expanded compared to previous studies have not been presented. Furthermore, if the research results include specific practical implications for the South African university organization, the academic and practical value of this study will be further enhanced. It is believed that the understanding of this study will be improved if the conclusion part is presented by dividing it into discussion, theoretical implications, and practical implications.

Reviewer 3 Report

Comments and Suggestions for Authors

Thank you for the opportunity to review the manuscript entitled “Workplace mental health status among academic staff: Psychological distress, burnout, and organisational culture at a South African University”. The study aims to assess the prevalence of psychological distress and burnout among academic staff, to explore organisational culture regarding mental health, and to identify staff preferences for mental health interventions within a historically disadvantaged South African university.

The manuscript makes a valuable contribution by providing institution-wide data on psychological distress, burnout, and organisational culture, and by identifying staff preferences for mental health interventions. The use of validated measures (GHQ-28, OLBI) alongside context-specific items is a notable strength, and the findings have both theoretical and applied implications for workplace mental health. However, several revisions are required before the manuscript is suitable for publication. These revisions primarily concern methodological reporting, theoretical framing, and editorial issues.

Abstract

1) The abstract is clear but could be more specific regarding demographics. Reporting the median age, standard deviation, and gender distribution would enhance transparency and align with APA reporting standards.

Introduction

2) The introduction provides a solid contextualisation of academic burnout and distress in the Global South. However, the framing could be strengthened by integrating more theoretical perspectives on organisational culture and psychological safety (e.g., Edmondson & Lei, 2014), given these are central to the findings.

3) The framing could be strengthened by expanding on the structural and systemic factors unique to historically disadvantaged institutions in South Africa and linking them explicitly to the study aims.

4) The framing could be strengthened by including more recent comparative international data on staff burnout/distress to highlight the gap this study fills. Indeed, burnout is a multi-component construct formed by different and complex size. It is also necessary to refer and comment on the authors' results in relation to the change that the theoretical framework on Burnout has undergone in recent years. Furthermore, it is necessary to integrate the literature review, also referring not only to your own national context, but also international literature. Here are some recent works that suit your theme and which I think may be useful for expanding and updating this section:

- Angelini et al. (2024). Engaged teachers and well-being: The mediating role of burnout dimensions. Health Psychology and Behavioral Medicine, 12(1), 2404507.

- Pyhältö et al. (2020). Teacher burnout profiles and proactive strategies. European Journal of Psychology of Education, 1-24.

- Schaufeli et al. (2020). Burnout Assessment Tool (BAT)—development, validity, and reliability. International Journal of Environmental Research and Public Health, 17(24), 9495.

Expanding the literature review beyond a national context by incorporating international perspectives would provide a more comprehensive understanding of the topic.

Methods

5) The demographic table is comprehensive, but much of the information is descriptive and not central to the study’s main findings. I suggest summarizing the key participant characteristics (e.g., median age, gender distribution, academic rank distribution) directly in the text, while removing redundant or unnecessary details from the table. This will improve readability and avoid overloading the Results section with background information.

6) Currently, the “Survey instrument” section is embedded within the Methods alongside participants and procedures. For clarity and alignment with standard reporting practices, the measurement tools should be presented in a distinct subsection entitled “Measures” or “Instruments.”

7) Furthermore, at present, it omits crucial psychometric details that are necessary for replication and for assessing measurement validity. A more rigorous structure would include:

- The full name of each instrument (with original developer in parentheses), followed by the translated/adapted version used.

- Whether a validated version in the local language was used; if not, a Confirmatory Factor Analysis (CFA) should be conducted and reported.

- Model fit indices (e.g., CFI, TLI, RMSEA, SRMR) for each multi-item measure.

- A short description of subscales, number of items, and an example item from each scale.

- Internal consistency indices (Cronbach’s alpha) for all subscales and full scales.

8) Without these details, the validity, reliability, and cultural applicability of the measures remain uncertain. If full reporting is not possible, these omissions must be explicitly acknowledged as limitations.

9) For the institutional culture and needs assessment sections (developed de novo), more detail is needed on item development and whether content validity was assessed by experts.

10) The ethical approval statements are adequate but could be rephrased into a concise, formal IRB declaration.

Results

11) Tables should be formatted according to APA guidelines (consistent decimal places, italicised statistical symbols, etc.).

12) Ensure clarity in reporting GHQ-28 thresholds, currently mild/moderate/severe categories differ from some published norms, so a rationale for adaptation should be emphasised.

13) The regression analyses could be reported with effect sizes (ORs, CIs) in a clearer summary table rather than spread across text.

Discussion

14) The discussion effectively situates the findings within the broader literature, but it would benefit from a deeper unpacking of the gender differences observed in the study. In particular, the authors could draw more explicitly on theories of work–family conflict and intersectional perspectives on women in academia. This would allow them to go beyond the South African literature they already cite and position their findings within a wider theoretical framework that highlights the compounded pressures women often face in higher education.

15) Another area that could be strengthened is the theoretical elaboration on psychological safety and stigma. Both concepts are central to the results presented, yet their role as mechanisms linking organisational culture and mental health outcomes could be articulated with greater clarity. Expanding this part of the discussion would not only provide stronger conceptual grounding but also help explain why participants reported low levels of disclosure and trust in their institution.

16) The dual academic–clinical roles of many participants also deserve more explicit attention. These roles, which combine the demands of teaching and research with clinical responsibilities, likely contribute to the elevated levels of burnout found in the sample. Highlighting this intersection would provide a more nuanced interpretation of the findings and situate them within the specific context of a health sciences university.

17) The discussion would also gain practical value from a stronger focus on interventions and policies that are feasible within resource-constrained environments. By pointing to concrete strategies that universities in low- and middle-income countries could realistically adopt, the paper would better bridge the gap between evidence and application, enhancing its relevance for policymakers and institutional leaders.

18) Turning to minor issues, there are several editorial and technical aspects that require attention. Statistical terms such as M, SD, p, r, and CI should be italicised consistently throughout the text, and redundant or unclear phrasing should be corrected—for example, the phrase “the survey was stuck to” should be rephrased for clarity. Minor typographical errors, such as “trillion lost,” should be revised to read “USD 1 trillion lost.” The reference list also needs careful editing, as some entries are incomplete, for instance, Goldberg and Williams (1988) is cited with “(No Title).” Finally, abbreviations should be introduced and defined consistently at first use, including key terms such as HDI and LMIC.

Conclusion

19) The conclusion is generally well aligned with the findings, but it could be made stronger by explicitly highlighting the novel contribution of the study. In particular, the authors should stress that this is the first institution-wide assessment of academic staff wellbeing at this setting and emphasise the implications of their results for the development of a context-specific, co-designed mental health policy. By framing the conclusion in this way, the paper would more clearly underline its originality and practical significance.

Round 2

Reviewer 2 Report

Comments and Suggestions for Authors

I believe the researchers have adequately enhanced the value of this study.

In my opinion, the concept has been faithfully reflected throughout the introduction, the definition, the contextual explanation, the gender studies, and the discussion of discrimination. In addition, hypotheses have been newly presented, and theories and prior studies supporting the relationship between variables have been strengthened.

In the conclusion, the connection with previous studies was strengthened. The discussion, comparison, and interpretation of various international and African studies were enhanced, and academic and practical implications were highlighted, including the JD-R model, the improvement of organizational culture, and the need for policy intervention.

Reviewer 3 Report

Comments and Suggestions for Authors

Thank you to the authors for addressing my suggestions. Aside from some minor editing adjustments and a few remaining issues (e.g., in the Instruments section, “Section A1” has been removed but the other section labels before the instrument names remain, these should be deleted since the instrument name alone is sufficient to indicate the sub-section), the manuscript is suitable for publication. Thank you.